# Evolving Pareto-Optimal Reasoning Paths in LLMs

**Jinuk Lee**
Department of Artificial Intelligence
Kyungpook National University
Daegu, South Korea
`dnr9333@knu.ac.kr`

## Abstract

Recent advancements in Large Language Models (LLMs) have demonstrated remarkable capabilities in mathematical reasoning. However, relying solely on greedy decoding or multi-agent frameworks often leads to suboptimal reasoning paths, particularly in complex, multi-step problems. In this paper, we propose a novel Self-Evolving Reasoning Framework that treats the generation of reasoning steps as a multi-objective optimization problem. Unlike traditional methods that maximize a scalar reward, our approach utilizes the Non-dominated Sorting Genetic Algorithm II (NSGA-II) to navigate trade-offs between conflicting objectives, such as solution accuracy, logical coherence, and computational efficiency (penalty minimization). The framework generates a population of reasoning paths and iteratively evolves them to approximate the Pareto-optimal front. Experimental results demonstrate that our framework significantly enhances the agent's robustness and adaptability in solving complex mathematical problems.

## 1 Introduction

Mathematical reasoning represents a fundamental challenge in artificial intelligence, demanding not only linguistic fluency but also precise logical deduction and multi-step planning. While the advent of Large Language Models (LLMs) has significantly advanced the state of the art, standard approaches relying on Chain-of-Thought (CoT) prompting or supervised fine-tuning often struggle with robustness in complex problem-solving scenarios. A critical limitation in current methodologies is the reliance on scalar verifiers Wang et al. (2023); Kirchner et al. (2024); Luo et al. (2024). Existing frameworks typically maximize a single scalar reward signal—primarily the accuracy of the final answer or language feedback—while largely neglecting the quality and efficiency of the reasoning process itself. Consequently, models frequently converge on solutions that are factually correct but logically brittle, computationally redundant, or derived through spurious correlations within a narrow search space.

The fundamental difficulty lies in the multi-dimensional nature of reasoning. An ideal mathematical agent should not merely derive a correct answer; it should do so by balancing conflicting objectives, such as maximizing logical rigor and axiomatic fidelity (reward) while minimizing computational overhead, reasoning redundancy, and heuristic bias (penalty). In many cases, a concise proof is preferred over a verbose one not just for efficiency, but because it often reflects a deeper understanding of the underlying mathematical structure. To navigate this complex landscape, a framework is required that can explicitly handle the trade-offs between these competing goals without reducing them to a monolithic scalar value.In this work, we propose a novel Self-Evolving Reasoning Framework that formulates the generation of mathematical proofs as a multi-objective optimization problem. Departing from standard greedy or sampling-based decoding, we integrate the Non-dominated Sorting Genetic Algorithm II (NSGA-II) into the agent's inference process. By treating each reasoning step as a genetic trait that can be mutated or recombined, the framework explores the solution space with a level of diversity that single-path agents cannot achieve. This approach allows the system to maintain a diverse population of reasoning paths, iteratively evolving them to approximate the Pareto-optimal front.By explicitly optimizing for both high rewards (correctness) and low penalties (inefficiency), the agent can identify solutions that are not only accurate but also structurally con-

cise and logically robust. This process is further augmented by a long-term memory module that stores elite reasoning patterns, allowing the agent to "learn how to learn" from its evolutionary history. This feedback loop ensures that the agent retains successful optimization strategies, effectively selecting optimal solutions from its own reasoning history. By refining these evolved patterns for future tasks, the system achieves a self-improving cycle, significantly enhancing its adaptability and performance on complex mathematical problems.

## 2 RELATED WORK

### 2.1 SELF-EVOLVING REASONING

Recent work has investigated self-evolving frameworks that enable language models and agents to iteratively improve their reasoning without external supervision. Early approaches leverage self-evaluation or self-refinement signals, allowing models to generate, assess, and revise their own outputs Hosseini et al. (2024); Wang et al. (2024); Kang et al. (2025). While effective, these methods typically rely on heuristic or scalar reward functions and lack explicit mechanisms for structured exploration.

A complementary line of research focuses on search-based self-evolving reasoning, particularly for mathematical problem solving. Methods based on Monte Carlo Tree Search (MCTS) explore reasoning trajectories and select high-quality solutions Guan et al. (2025); Zhang et al. (2024a;b). These approaches improve correctness by expanding the search space but generally optimize a single objective and discard intermediate reasoning patterns once inference terminates.

Population-based optimization has been explored through genetic and evolutionary algorithms. Works such as EvoAgent, Evolving Deeper LLM Thinking, and EvoPrompt apply mutation and selection over prompts or reasoning structures Yuan et al. (2025a); Lee et al. (2025); Guo et al. (2024). Although these methods enhance exploration beyond greedy decoding, they typically adopt scalar fitness functions, limiting their ability to model trade-offs among competing reasoning objectives.

Self-evolving paradigms have also been applied in domain-specific settings. In mathematical discovery, evolutionary systems enable large-scale exploration of conjectures and solution patterns Novikov et al. (2025); Georgiev et al. (2025). In code evolution, iterative modification and evaluation of programs improves correctness and performance Assumpção et al. (2026); Hu et al. (2026). Related ideas extend to GUI understanding, where agents refine perception-action loops through iterative feedback Wang et al. (2025); Yuan et al. (2025b). Despite their success, these systems are often domain-dependent and rely on handcrafted evaluation criteria.

Recent surveys highlight that a central open challenge in self-evolving AI is the absence of principled multi-objective optimization frameworks for reasoning evolution Fang et al. (2025); ang Gao et al. (2026). In contrast to prior work, we formulate reasoning evolution as a multi-objective optimization problem and employ NSGA-II to evolve a population of reasoning paths, explicitly balancing accuracy, coherence, and computational efficiency while retaining high-quality reasoning patterns via long-term memory.

### 2.2 PROBLEM SETUP AND PRELIMINARIES

**Search Space.** Let $\mathcal{S}$ denote the search space of candidate mathematical solutions or algorithms, where each candidate $p \in \mathcal{S}$ can be represented as a source program, an abstract syntax tree (AST), or a proof object:

$$\mathcal{S} = \{p \mid p \text{ is a syntactically valid candidate solution for problems in } \mathcal{D}_{\text{prob}}\}.$$

Feasible candidates must satisfy a constraint $C(p)$ (e.g., type-checking, compilation, and sandbox safety).

**Evaluator and Objectives.** Let $E : \mathcal{S} \to \mathbb{R}^k$ be an evaluator that returns a vector of objectives:

$$E(p) = \big(\text{correctness}(p), \ \text{partial\_score}(p), \ \text{runtime}(p), \ \text{proof\_length}(p)\big).$$

Correctness may be a binary indicator or a fractional score based on test cases or formal verification. The optimization target is a multi-objective Pareto front, or a scalarized utility derived from $E(p)$.

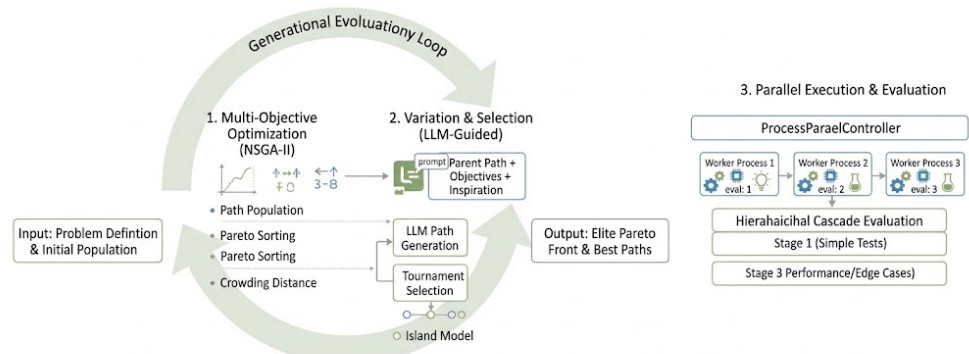

Figure 1: Overview of the complete architecture for LLM-guided evolution. The system operates in a generational evolutionary loop consisting of three main phases:

**Evolutionary Search Framework.** We perform evolutionary search over $\mathcal{S}$ with population $P_t$. Mutation operators include LLM-guided rewrites $G_{\text{LLM}}(\cdot \mid p)$ and domain-preserving symbolic transforms; crossover is optional. Selection uses multi-objective criteria (Pareto dominance, hypervolume, or MAP-Elites-style archives to preserve behavioral diversity). The algorithm iteratively updates $P_t \to P_{t+1}$ subject to a compute and LLM inference budget $B$.

**Practical Constraints.** We enforce a finite round budget $L$, an overall LLM and compute budget $B$, sandboxed execution for safety, and repeated measurements to handle evaluation noise. Reproducibility is ensured via fixed random seeds and released configurations and prompts where applicable.

## 3 Methodology

The proposed framework treats the generation of mathematical proofs as an evolutionary search over the space of reasoning trajectories. By utilizing NSGA-II, we maintain a population of diverse reasoning paths and iteratively refine them to balance correctness with structural elegance and logical rigor.

### 3.1 Reasoning Path Representation and Objectives

Let $\mathcal{R}$ denote the search space of candidate reasoning paths for a given mathematical problem. A single reasoning path $p \in \mathcal{R}$ is defined as a sequence of logical steps $\{s_1, s_2, \ldots, s_n\}$, where each step $s_i$ represents an intermediate deduction, calculation, or application of a mathematical theorem. The fitness of a reasoning path is evaluated through a multi-objective vector $\mathbf{v}(p) = [f_{acc}(p), f_{coh}(p), f_{eff}(p)]$:

- **Accuracy** ($f_{acc}$): A measure of whether the final derivation reaches the ground-truth solution, verified through symbolic solvers or formal verification tools.
- **Logical Coherence** ($f_{coh}$): An assessment of the transitions between $s_i$ and $s_{i+1}$, ensuring that each step follows logically from the previous ones without unjustified heuristic leaps.
- **Efficiency** ($f_{eff}$): A penalty-based objective that favors conciseness. It minimizes redundant steps or reasoning loops that do not contribute to the final proof, promoting "elegant" solutions.

### 3.2 Multi-Objective Evolution via NSGA-II

To navigate the trade-offs between these potentially conflicting objectives, we employ the Non-dominated Sorting Genetic Algorithm II (NSGA-II) to maintain a population $P$ of $N$ reasoning

paths. At each generation, paths are partitioned into hierarchical Pareto fronts $(\mathcal{F}_1, \mathcal{F}_2, \dots)$ based on the dominance relation. A path $p_1$ is said to dominate $p_2$ ($p_1 \prec p_2$) if it is superior in at least one objective and no worse in all others.

To prevent the model from converging on a single reasoning strategy, we apply *crowding distance* $d_i$ to reward paths that explore unique regions of the logical search space:

$$d_i = \sum_{j \in \{acc, coh, eff\}} \frac{f_j(p_{i+1}) - f_j(p_{i-1})}{f_j^{\max} - f_j^{\min}}. \tag{1}$$

This ensures that the framework explores diverse mathematical approaches (e.g., geometric vs. algebraic) simultaneously by maintaining behavioral diversity within the population.

### 3.3 LLM-Guided Logical Variation

Unlike traditional evolutionary algorithms that use stochastic mutations, our framework utilizes a Large Language Model (LLM) as a sophisticated genetic operator $G_{\text{LLM}}$.

1. **Parent Selection:** Two paths are selected via binary tournament based on their Pareto rank and crowding distance.

2. **Logical Recombination (Crossover):** The LLM is prompted to identify successful sub-proofs from both parents and integrate them into a more robust offspring, effectively combining different logical insights.

3. **Step Refinement (Mutation):** The LLM identifies a specific weak link or redundant segment in a path and regenerates it with higher precision, guided by textual feedback from the evaluation stage.

### 3.4 Parallel Verification Architecture

To handle the computational load of evaluating multiple reasoning trajectories, the system employs a distributed architecture. Each candidate path is evaluated in an isolated environment across three specialized modules:

- **Verifiers:** Check the validity of algebraic manipulations and numerical calculations.
- **Critics:** LLM-based agents that scrutinize the axiomatic flow and identify logical fallacies.
- **Analyzers:** Measure the information density and structural conciseness of the reasoning path.

## 4 Experiments

In this section, we conduct experiment to evaluate the efficacy of our Self-Evolving Reasoning Framework using Gemini-3.1-Pro. Our primary goal is to determine whether the integration of NSGA-II allows an LLM-based agent to discover optimal or near-optimal solutions in combinatorial and geometric search spaces that are traditionally difficult for standard greedy decoding.

We have selected three classic challenges in geometric optimization: the Moving Sofa Problem, the Erdős–Szekeres Happy Ending Problem, and Packing in a Dilate. These problems were chosen because they require a unique combination of continuous optimization and discrete logical constraints. For each task, the framework was initialized with a diverse population of candidate solutions, and the evolutionary process was carried out over multiple generations to observe the convergence toward the Pareto-optimal front. The evaluation metrics include:

### 4.1 Problem Setup

**(1) Moving Sofa Problem.** This problem encompasses a class of geometric maximization problems that test the limits of rigid shapes moving through constrained corridors. The objective is to determine the largest area of a connected planar shape (a "sofa") that can continuously pass through an L-shaped corridor of unit width.

---

**Algorithm 1** Self-Evolving Reasoning Framework via NSGA-II

---

**Require:** Problem domain $\mathcal{D}$, Population size $N$, Max generations $G$, LLM engine $G_{\text{LLM}}$
**Ensure:** Pareto-optimal reasoning paths $\mathcal{F}_1$

1: $P_0 \leftarrow \text{InitializePopulation}(\mathcal{D}, N)$        ▷ Initial reasoning paths from LLM
2: $E_0 \leftarrow \text{Evaluate}(P_0)$        ▷ Measure Accuracy, Coherence, Efficiency
3: $\mathcal{F} \leftarrow \text{NonDominatedSort}(P_0, E_0)$        ▷ Rank-based partitioning **for** $t = 1$ **to** $G$ **do**
4:
       $Q_t \leftarrow \emptyset$ **while** $|Q_t| < N$ **do**
5:
       $p_1, p_2 \leftarrow \text{TournamentSelection}(P_{t-1}, \mathcal{F})$
6: $p_{new} \leftarrow G_{\text{LLM}}(\text{variation} \mid p_1, p_2, \text{feedback})$        ▷ LLM-guided refinement
7: $Q_t \leftarrow Q_t \cup \{p_{new}\}$
8:
9: $R_t \leftarrow P_{t-1} \cup Q_t$        ▷ Combine parents and offspring
10: $E_t \leftarrow \text{Evaluate}(R_t)$
11: $\mathcal{F}_1, \mathcal{F}_2, \cdots \leftarrow \text{NonDominatedSort}(R_t, E_t)$
12: $P_t \leftarrow \emptyset, \quad i \leftarrow 1$ **while** $|P_t| + |\mathcal{F}_i| \leq N$ **do**
13:
       $P_t \leftarrow P_t \cup \mathcal{F}_i$
14: $i \leftarrow i + 1$
15:
16: $P_t \leftarrow P_t \cup \text{SelectByCrowdingDistance}(\mathcal{F}_i, N - |P_t|)$
17:
18: **return** $\mathcal{F}_1$

---

**(2) Erdős–Szekeres Happy Ending Problem.** This problem asks for the minimum number of points required in the plane to guarantee the existence of a convex $k$-gon.

**(3) Packing in a Dilate.** This problem defines a geometric scaling challenge focused on efficient spatial arrangement. The objective is to determine $C(n, P)$, the minimum scale factor $s$ such that $n$ non-overlapping copies of a given planar shape $P$ can be contained within a scaled copy $sP$.

## 4.2 MOVING SOFA PROBLEM

The classic moving sofa problem, posed by Moser (1966), asks for the maximum area of a connected planar region that can pass through a unit-width L-shaped corridor. Let $C$ denote this maximum area. Gerver (1992) constructed a sofa achieving area $C \geq 2.2195\ldots$; this value has since been proven optimal, establishing

$$C = 2.2195\ldots$$

Our framework was applied to this problem and successfully reproduced this known optimal value.

## 4.3 ERDŐS–SZEKERES HAPPY ENDING PROBLEM

Given a finite set of points in the plane in general position, the *happy ending problem* asks how many points suffice to guarantee the existence of a convex $k$-gon. Erdős and Szekeres showed that

$$2^{k-2} + 1 \leq C(k) \leq 2^{k+O(\sqrt{k \log k})},$$

with the lower bound construction and the upper bound established in Holmsen et al. (2020). For small values of $k$, the exact answer is known: Klein showed $C(4) = 5$, while $C(5) = 9$ and $C(6) = 17$ were resolved through substantial computational effort using SAT solvers

To probe the lower bound for $k \leq 8$, we ran our framework with the convex $k$-gon count as the objective to minimize. Degenerate configurations—those with nearly coincident points or near-collinear triples—were penalized with a score of negative infinity to enforce general position. Across all values of $k$, our framework reliably produced $2^{k-2}$-point configurations containing no convex $k$-gon, consistent with the known extremal constructions. Pushing to $2^{k-2} + 1$ points proved unsuccessful, leaving the lower bound unimproved.

### 4.4 PACKING IN A DILATE

The packing-in-a-dilate problem asks for the smallest scale factor $s$ such that $n$ interior-disjoint copies of a geometric body $P \subset \mathbb{R}^d$ can be placed inside a scaled copy $sP$. Let $C(n, P)$ denote this minimal scale. A volume argument gives the universal lower bound

$$C(n, P) \geq n^{1/d}.$$

More generally, if $\delta^*(P)$ denotes the optimal packing density of $P$, then asymptotically

$$C(n, P) \geq \left(\frac{n}{\delta^*(P)}\right)^{1/d}(1 - o(1)).$$

Conversely, lattice packing constructions yield

$$C(n, P) \leq \left(\frac{n}{\delta_L(P)}\right)^{1/d} + O\left(n^{(d-1)/d^2}\right),$$

where $\delta_L(P)$ is the optimal lattice packing density. Although the asymptotic behavior of $C(n, P)$ is governed by packing density considerations, finite-$n$ instances exhibit significant boundary effects. Our framework successfully generated geometrically diverse configurations whose achieved scale factors closely approach the theoretical lower bounds, demonstrating its flexibility in handling geometric packing optimization tasks.

## 5 CONCLUSION

This paper introduced a Self-Evolving Reasoning Framework that treats LLM reasoning path generation as a multi-objective optimization problem. By integrating NSGA-II, we navigate trade-offs between accuracy, logical coherence, and computational efficiency. Our results on challenges, including the Moving Sofa and Happy Ending problems, demonstrate the framework's ability to efficiently explore combinatorial search spaces and recover known optimal or extremal configurations.

A primary limitation of this study is the restricted number of experiments conducted and not yet demonstrate research-level mathematical maturity due to absence of rigorous mathematical verification by domain experts. While the results are promising, more extensive large-scale benchmarking is required to fully validate statistical robustness across diverse problem domains.

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
