# OpenReview forum: "Evolving Pareto-Optimal Reasoning Paths in LLMs"
_mathai.club/MathAI/2026/Conference — 2026 Oral_

### Official Review · Reviewer_a9nC · 2026-03-13
**A novel evolutionary framework for LLM reasoning with incomplete experimental analysis**

**Rating:** 6
**Confidence:** 4

**Review:**

Summary
This paper proposes a Self-Evolving Reasoning Framework for LLM-based mathematical problem solving. The core idea is to treat the generation of reasoning paths as a multi-objective optimization problem, employing NSGA-II to evolve a population of candidate reasoning paths across three objectives: accuracy, logical coherence, and efficiency. Experiments are conducted on three open mathematical problems: the Moving Sofa Problem, the Erdős–Szekeres Happy Ending Problem, and Packing in a Dilate.

Strengths
1. Conceptual novelty. The idea of applying multi-objective Pareto optimization to the reasoning path search space is a legitimate and interesting extension to the growing body of work on LLM with evolutionary algorithms.
2. Relevant framing. The paper correctly identifies a known weakness in the field: most existing work collapses multi-dimensional reasoning quality into a single reward signal. The three-objective decomposition is conceptually sound and maps onto real requirements.

Weaknesses
1. Incompleteness. The experimental section lacks a complete description of the validation metrics employed.
2. Scalability problem. No runtime analysis of the proposed method is provided, which raises concerns regarding its scalability to larger problem settings.
3. Reproducement. The underlying LLM configuration is not specified, making the reported experiments impossible to reproduce.
4. Experimental results. The set of validation tasks is limited, and the empirical evaluation presented for them is not supported by quantitative metrics. Moreover, there is no comparison with any baseline, so it seems impossible to estimate an efficiency of proposed method.

Taken together, these issues cast doubt on the practical relevance and overall significance of the study, given its seeming limitations. It would be beneficial if the authors provided evidence demonstrating the significance of the proposed approach.

---

### Official Review · Reviewer_83Dg · 2026-03-13
**Pareto-based evolutionary reasoning for LLMs: an interesting direction with limited experimental grounding**

**Rating:** 5
**Confidence:** 3

**Review:**

Contribution.
The paper proposes a framework that formulates reasoning path generation for mathematical problem solving as a multi-objective evolutionary optimization task. The use of NSGA-II to evolve candidate reasoning trajectories under objectives related to correctness, coherence, and efficiency is an interesting conceptual direction that aligns with recent trends combining LLM reasoning with search-based methods.

Positive aspects.
The idea of explicitly modelling trade-offs between different reasoning quality dimensions is well motivated. The overall system design is understandable at a high level, and the chosen mathematical problems illustrate the intended reasoning workflow.

Main limitations.
The empirical evaluation is narrow and largely qualitative, making it difficult to assess the actual performance benefits of the proposed approach. The paper does not include comparisons with baseline reasoning strategies or quantitative metrics that would allow estimating effectiveness. Important implementation details such as model configuration, search budget, and runtime characteristics are also missing, which limits reproducibility and raises concerns about scalability.

Overall.
The work presents a promising conceptual framework, but stronger experimental validation and clearer reporting would be necessary to demonstrate its practical significance.

---

### Official Review · Reviewer_ZLHv · 2026-03-13
**There are problems in the "Evolving Pareto-Optimal Reasoning Paths in LLMs" paper**

**Rating:** 5
**Confidence:** 3

**Review:**

There are problems in the "Evolving Pareto-Optimal Reasoning Paths in LLMs" paper

This paper is devoted to solution of such important task as improving mathematical reasoning in Large Language Models. Athors try to solve this problem using Self-Evolving Reasoning Framework that treats the generation of reasoning as a multi-objective optimization problem.

This paper has the following disadvantages:
1) Authors have described that logical coherence as one of the objective for optimization but authors have not considered how to formalize logical coherence.
2) It is necessary to correct "?" to appropriate reference on description of bounds of happy ending problem.

---

### Decision · Program_Chairs · 2026-03-14

**Decision:**

Accept (Oral)

**Comment:**

Dear Author(s),

On behalf of the Program Committee of the International Conference on Mathematics of Artificial Intelligence (MathAI 2026), we are pleased to inform you that your paper has been accepted for an oral presentation at MathAI 2026.

Your paper was evaluated through a rigorous two-stage review process involving both automated screening and expert review by members of the Program Committee. The reviewers recognized the quality and contribution of your work.

Presentation details:

- Format: Oral presentation (15–20 minutes + 5 minutes Q&A)
- Mode: You may present either in person (offline) at the conference venue in Sirius, Russia, or remotely via Zoom. Please indicate your preferred mode when confirming your participation.
- Conference dates: Marh 30 - April 3, 2026
- Website: https://mathai.club

Next steps:

1. Please confirm your participation and presentation mode by replying to this email mathai.club@yandex.ru no later than March 15, 2026 18:00 Moscow time.
2. If you plan to attend in person, the organizing committee will provide accommodation details separately.
3. Please prepare your final camera-ready manuscript according to the formatting guidelines available at https://mathai.club and upload it to OpenReview by March 15, 2026 18:00 Moscow time.

Should you have any questions regarding the program, logistics, or your presentation slot, please do not hesitate to contact us.

We look forward to your contribution to MathAI 2026.

With kind regards,

MathAI 2026 Program Committee
International Conference on Mathematics of Artificial Intelligence
https://mathai.club
OpenReview: https://openreview.net/group?id=mathai.club/MathAI/2026/Conference
Telegram: https://t.me/MathAI_club
Email: mathai.club@yandex.ru